# SuperMag: Vision-based Tactile Data Guided High-resolution Tactile Shape Reconstruction for Magnetic Tactile Sensors

Peiyao Hou[1,2*], Danning Sun[1*], Meng Wang[2], Yuzhe Huang[1,2],
Zeyu Zhang[2], Hangxin Liu[2], Wanlin Li[2†], Ziyuan Jiao[2†]

*Abstract*— Magnetic-based tactile sensors (MBTS) combine the advantages of compact design and high-frequency operation but suffer from limited spatial resolution due to their sparse taxel arrays. This paper proposes SuperMag, a tactile shape reconstruction method that addresses this limitation by leveraging high-resolution vision-based tactile sensor (VBTS) data to supervise MBTS super-resolution. Co-designed, open-source VBTS and MBTS with identical contact modules enable synchronized data collection of high-resolution shapes and magnetic signals via a symmetric calibration setup. We frame tactile shape reconstruction as a conditional generative problem, employing a conditional variational auto-encoder to infer high-resolution shapes from low-resolution MBTS inputs. The MBTS achieves a sampling frequency of 125 Hz, whereas the shape reconstruction sustains an inference time within 2.5 ms. This cross-modality synergy advances tactile perception of the MBTS, potentially unlocking its new capabilities in high-precision robotic tasks.

## I. INTRODUCTION

Among current tactile sensing techniques, Magnetic-based Tactile Sensors (MBTS) [1–7] offer advantages such as compact and simple designs, high response frequencies (> 100 Hz), multi-axis force detection, and cost-effectiveness. A key limitation of MBTS, shared with other non-vision-based method [8], is their taxel-array configuration, which restricts spatial resolution due to the physical space occupied by each sensing element. This limitation impedes their performance in applications requiring fine-grained tactile perception.

Vision-based Tactile Sensors (VBTS) [9–21], offer a promising solution to these limitations. Despite their bulky form factor and lower frequencies (30-60 Hz), VBTS inherently achieve high-resolution shape reconstruction through direct visual feedback by leveraging learned pixel-level depth mappings from minimal training data [22, 23]. The complementary strengths and weaknesses of MBTS and VBTS suggest a synergistic potential. We propose that the easily acquired tactile data from VBTS—which encapsulate fine geometric and textural details—could serve as supervisory signals to guide MBTS in reconstructing high-resolution shapes. By integrating cross-modal learning frameworks, the high-resolution priors captured by VBTS could enable MBTS to surpass their physical resolution limits, bridging the gap between sparse tactile data acquisition and dense, accurate shape estimation.

* Peiyao Hou and Danning Sun contributed equally to this work. This work was conducted during Peiyao Hou's internship at the Beijing Institute for General Artificial Intelligence (BIGAI). † Corresponding authors. [1] Department of Automation, Beihang University. [2] State Key Laboratory of General Artificial Intelligence, BIGAI, Beijing, China.

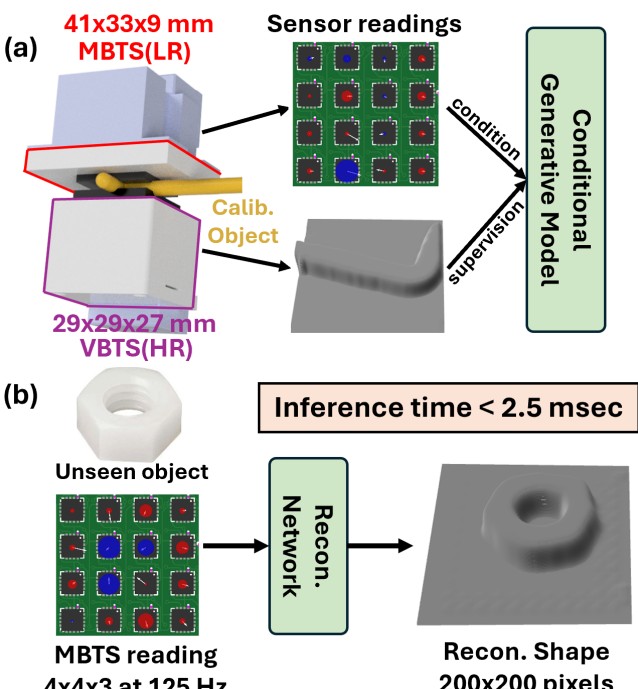

Fig. 1: **SuperMag: High-resolution Tactile Shape Reconstruction for Magnetic-based Tactile Sensors (MBTS) with Vision-based Tactile Sensors (VBTS) data.** (a) Training: High-Resolution (HR) VBTS [24] depth images serve as supervisory signals to guide Low-Resolution (LR) MBTS [6] in reconstructing high-resolution tactile shapes of the object. (b) Inference: Sparse MBTS data are used to reconstruct the tactile shape of an unseen test object.

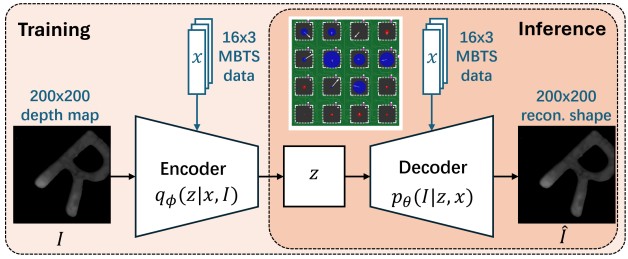

Fig. 2: Network architecture of SuperMag.

In this work, we propose SuperMag, a tactile shape reconstruction method that leverages high-resolution tactile data from VBTS [24] to guide the super-resolution of low-resolution MBTS signals [6], enabling high-spatial-resolution shape reconstruction at high operational frequencies. We frame the reconstruction task as a conditional generative problem, where high-resolution depth maps are inferred

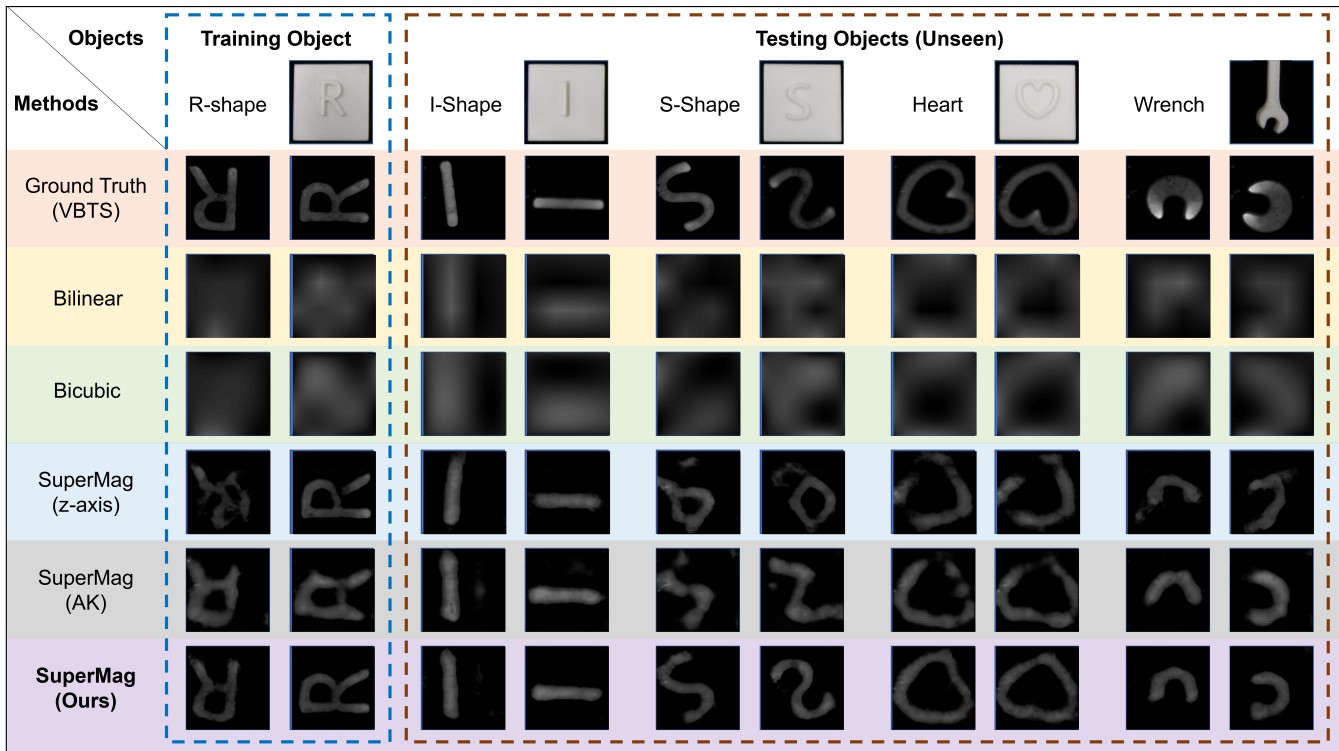

Fig. 3: **Shape reconstruction results for ground truth, baselines, and SuperMag on seen training object and unseen testing objects.** The ground truth is from the vision-based tactile sensor (VBTS). Baselines are Bilinear and Bicubic interpolation of z-axis magnetic-based tactile sensor (MBTS) data. SuperMag (z-axis) is trained on R-shape z-axis MBTS data; SuperMag (AK) is trained on Allen key (AK) MBTS data, noted that R-shape is an unseen object for SuperMag (AK); SuperMag is trained on both Allen key and R-shape MBTS data.

from sparse MBTS readings using a conditional variational autoencoder (CVAE). The resulting dataset comprises 2025 pairs of tactile readings for the Allen key and 2025 pairs for the letter "R". Experiments show that SuperMag outperforms baseline methods in both quantitative metrics and qualitative evaluations, achieving high-resolution shape reconstruction with an inference time of 2.5 ms per reading.

TABLE I: Comparison of SuperMag against baselines.

| Method Name | FID↓ | PSNR[dB]↑ | SSIM↑ |
|---|---|---|---|
| Bilinear | 402.63 | 8.03 ± 1.78 | 0.10 ± 0.04 |
| Bicubic | 309.10 | 6.75 ± 1.60 | 0.10 ± 0.04 |
| SuperMag (z-axis) | 234.16 | 20.86 ± 1.93 | 0.69 ± 0.08 |
| SuperMag (AK) | 213.43 | 22.36 ± 2.32 | 0.65 ± 0.07 |
| SuperMag (Ours) | **210.10** | **24.24** ± 2.88 | **0.78** ± 0.06 |

## II. DISCUSSION & CONCLUSION

This work presents SuperMag, a tactile shape reconstruction method that enables super-resolution of Magnetic-based Tactile Sensors (MBTS) using Vision-based Tactile Sensors (VBTS) data. Leveraging co-designed sensors with identical contact modules and a symmetric calibration setup, we train a conditional variational autoencoder (CVAE) to infer high-resolution tactile shapes from low-resolution MBTS inputs. SuperMag reconstructs $200 \times 200$ pixel shapes from $4 \times 4 \times 3$ taxel arrays, outperforming baselines in Frechet Inception

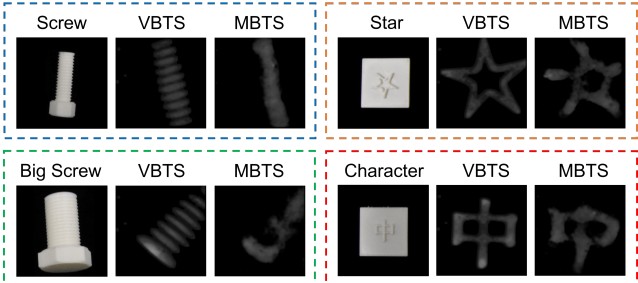

Fig. 4: Example results of SuperMag shape reconstruction for unseen objects with fine texture details.

Distance (FID), Peak-Signal-to-Noise Ratio (PSNR), and Structural Similarity (SSIM), while operating at a 95 Hz.

However, several limitations remain. First, the proposed method is currently constrained to MBTS sensors equipped with contact modules that match the dimensions and silicone material of the VBTS. Additionally, the use of MBTS may be unsuitable for grasping magnetizable materials. Future work will investigate the transferability of the approach across a broader range of taxel-based sensors. Second, while SuperMag excels at shape contour reconstruction, its ability to recover fine details remains inferior to VBTS (see Fig. 4), requiring further refinement. Finally, the inherent limitation of VBTS in detecting large planar surfaces impacts MBTS performance, necessitating additional research. These advancements aim to further bridge the gap between high-frequency and high-resolution tactile sensing for robotics.

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
