# OpenReview forum: "SuperMag: Vision-based Tactile Data Guided High-resolution Tactile Shape Reconstruction for Magnetic Tactile Sensors"
_IEEE.org/IROS/2025/Workshop/Tactile_Sensing — IROS 2025 Workshop Tactile Sensing OralPoster_

### Official Review · Reviewer_6uBJ · 2025-09-13
**Interesting work and sound approach (with suggestion to include dataset details)**

**Rating:** 8
**Confidence:** 5

**Review:**

This paper describes a method for reconstructing high-resolution tactile shapes (200x200) from a sparse magnetic sensor array (4x4). The authors use a conditional variational autoencoder to map sparse inputs to high-resolution images, with ground-truth data provided by a vision-based tactile sensor. The evaluation shows the method achieves lower FID and higher PSNR and SSIM scores compared to the baselines, indicating its potential for high-frequency and high-resolution robotic sensing applications. My only suggestion is to include the size of the datasets used for training and testing for clarity.

---

### Official Review · Reviewer_unMk · 2025-09-13
**An interesting work**

**Rating:** 7
**Confidence:** 4

**Review:**

This paper proposes a novel method for reconstructing high-resolution tactile image from MBTS data. There are some suggestion for this work: 1. To better illustrate the work, it is recommended to supplement detailed information about the dataset. Can the data used be open-sourced? 2. Could the author provide more details about the Network architecture of SuperMag and the selection of hyperparameters?